# Self-blaming as a barrier to lung cancer screening and smoking cessation programs in Italy. A qualitative study

**Luca Ghirotto** [1]*, **Eugenio Paci**[2], **Claudia Bricci**[2], **Silvia Marini**[2], **Valentina Bessi**[3], **Matías Eduardo Díaz Crescitelli**[1], **Ermanno Rondini**[4], **Francesco Pistelli**[3], **Giuseppe Gorini**[5], **Sandra Bosi**[4], **Paolo Giorgi Rossi**[6], **the Working Group**¶

1 Qualitative Research Unit, Azienda USL-IRCCS di Reggio Emilia, Reggio Emilia, Italy, 2 Italian League Against Cancer (LILT), Florence, Italy, 3 Pulmonary Unit, Cardiothoracic and Vascular Department, Pisa University Hospital, Pisa, Italy, 4 Italian League Against Cancer (LILT), Reggio Emilia, Italy, 5 Institute for Cancer Research, Prevention and Clinical Network (ISPRO), Florence, Italy, 6 Epidemiology Unit, Azienda USL-IRCCS di Reggio Emilia, Reggio Emilia, Italy

¶ Membership of the Working Group is provided in the Acknowledgments.
* luca.ghirotto@ausl.re.it

**Data Availability Statement:** The qualitative data underlying the findings of this study consist of interview and focus group transcripts, which inherently contain personal and sensitive

## Abstract

### Background

Lung cancer screening (LCS) combined with smoking cessation programs is a critical strategy for reducing lung cancer mortality. Understanding the perspectives of cigarette users and former ones on these interventions is essential for enhancing their acceptability and effectiveness. This study aimed to explore, in Italy, the perceptions and experiences of individuals eligible for LCS within the context of a smoking cessation program.

### Methods and findings

This multicenter qualitative study was conducted in two Italian regions as part of a larger project the Italian League against Cancer promoted. Using purposive sampling, we included (a) cigarette users and former ones who participated in an Italian trial, ITALUNG study, and (b) cigarette users who had been offered individual or group smoking cessation interventions and were theoretically eligible for screening in the following years (aged 50–70, ≥15 pack-years). Data were collected through open-ended semi-structured interviews and focus group meetings and analyzed using reflexive thematic analysis. The data analysis yielded six themes covering participants' views on the interactions between the two types of interventions (screening and smoking cessation program). Across their data, we generated the following themes: (i) depreciation and fatalism toward the risk of smoking, (ii) self-blaming and ethicality, (iii) ambivalent impact of the screening on smoking, (iv) LCS-related information and concerns, (v) teachable and motivating moments, and (vi) non-stigmatizing communication and testimony by professionals.

information. These transcripts were collected following ethical guidelines and research protocols approved by the ethics committee. Participants did not consent to publicly sharing their data, and public deposition would compromise their privacy and confidentiality, mainly since the data includes contextual information that could indirectly identify participants. The data are in Italian, which may limit accessibility to an international audience. However, all key data and information necessary to understand the findings of this study are included within the manuscript itself. This consists of the thematic analysis and illustrative quotes that provide transparency and rigor to the findings. The corresponding author may provide further anonymized details or summaries of the data upon reasonable request, ensuring that confidentiality is maintained and ethical guidelines are upheld. Researchers may contact the corresponding author (LG) for inquiries regarding the data. Furthermore, data access requests can be directed to Dr. Elisa Mazzini (elisa.mazzini@ausl.re.it), the Director of the Infrastructure for Research and Statistics at the Azienda USL-IRCCS of Reggio Emilia (Italy), who has been authorized by the General Director of the Local Health Authority to oversee matters related to Institutional Research.

**Funding:** This study contributed, without receiving funding, to the Italian Ministry of Health Pilot project "Progetto Pilota di un programma di screening per il tumore polmonare integrato con la cessazione del fumo: percorsi, selezione dei soggetti e protocolli diagnostici, in vista di una valutazione HTA". The study was partially funded by Emilia Romagna Regional Health Authority DGR n° 1800/2020. The research project "Analisi dei meccanismi d'azione psico-comportamentali con cui la proposta di adesione alla TAC spirale agisce sulle abitudini tabagiche e sulla disassuefazione al fumo" has been supported by a '5x1000' national grant from the Italian League against Cancer (LILT-Rome). This study was partially supported by the Italian Ministry of Health – Ricerca Corrente Annual Program 2025.

**Competing interests:** The authors have declared that no competing interests exist.

## Conclusions

Our study underscores the importance of avoiding stigma and respecting the dignity of cigarette users in implementing LCS and smoking cessation programs. Clear communication and supportive interactions with healthcare providers are crucial for enhancing the acceptability and effectiveness of these interventions. Future research should focus on quantifying these findings and exploring additional factors influencing the acceptability and effectiveness of combined LCS and smoking cessation programs.

## Introduction

Lung Cancer (LC) is the first cancer cause of death in Europe [1]. Smoking is the most decisive risk factor and has the most considerable attributable fraction in most countries [2, 3]. This makes LC a largely preventable disease through smoking cessation interventions and policies to reduce smoking habits [4, 5].

Besides, smoking cessation interventions have demonstrated effectiveness in facilitating smoking cessation, subsequently reducing lung cancer, cardiovascular diseases, and other smoking-related health issues [6]. In recent decades, there has been a noticeable reduction in age-adjusted incidence rates of lung cancer [7]. Despite this, due to population aging, the overall burden of lung cancer is decreasing slowly. Early diagnosis has the potential to lower disease fatality rates [8–10], although attempts in the 1970s were largely unsuccessful [11]. In 1999, Low Dose Computed Tomography (LDCT) screening was proposed to detect lung cancer at earlier stages. However, the efficacy of this approach required validation through randomized screening trials (RST) [12]. In 2011, the first positive results of the largest RST came from the U.S. National Lung Screening Trial [13] and other European studies, including three Italian ones (ITALUNG, DANTE, and MILD) [14–16]. After some years, data from the Belgian-Dutch NELSON study confirmed the reduction in lung cancer mortality [17]. According to those trials, LCS can reduce lung cancer mortality by using LDCT screening tests at annual intervals and high quality. LDCT screening allows the early detection of lung nodules before clinical symptom occurrence in high-risk subjects [9–12]. Most guidelines identify cigarette users (CUs) or former CUs with a smoking history of 25/30 packs/year or more, aged 55 to 74/80, as the primary target.

Despite the evidence of mortality reduction, the intervention's acceptability was low in the USA, where programs have been rapidly recommended and implemented since 2013 [18]. The European Council only recently recommended implementing LDCT screening in pilot projects in European countries. Modeling studies suggested that incorporating smoking cessation treatments into LCS programs enhances lung cancer mortality reduction achieved through screening alone [19–22]. Notably, even a moderately effective one-time smoking cessation intervention can result in increased life-years gained comparable to screening alone, as smoking cessation doesn't just impact lung cancer mortality but mortality from various causes [21]. Therefore, there is an agreement on the opportunity to combine both primary prevention (encouraging smoking cessation) and secondary prevention (lung cancer screening) to address the issue of smoking-related diseases comprehensively [19, 20].

However, the multi-component prevention proposal introduces additional complexities to its acceptability [18, 23]. Worldwide, LDCT screening from the perspectives of

future users has been explored whenever it was implemented [24–32]. To better understand the use of LDCT screening and its acceptability, forward-looking research proved essential to inform policies at the country level [33]. Gaining insights into how individuals perceive this combination can illuminate the intricate elements that impact healthcare choices. This is paramount since primary care and screening intersect with individual beliefs, cultural backgrounds, and emotional factors. Clarifying the psycho-behavioral mechanisms that determine the interactions between the two kinds of interventions (screening and smoking cessation program) is crucial in designing the potential implementation of LDCT screening hand in hand with smoking cessation interventions [34, 35].

Qualitative research enquires about participants' narratives to provide insights into beliefs, perceptions, and perspectives, which are invaluable for designing interventions that resonate with patients' real-world experiences [36]. The results of previously published qualitative studies indicate the need to consider the specific needs of individuals who decide to embark on a smoking cessation journey [24, 30–32, 37], given the uniqueness of the target population [26]. Those studies reveal that individuals considering smoking cessation and LCS face a range of emotional, cognitive, and practical challenges that influence their decision-making and participation [31]. Fear of discovering lung cancer often leads to knowledge avoidance, with individuals expressing reluctance to learn about their health status due to the perceived inevitability of invasive treatments or terminal outcomes [26]. Emotional responses such as shame, regret, and low self-esteem stemming from smoking further complicate decision-making, as many view LCS as a corrective measure rather than a preventive one [26, 28]. Misunderstandings about the purpose and benefits of screening, including the mistaken belief that it provides a general evaluation of lung health, create unrealistic expectations.

Additionally, skepticism about the effectiveness of early detection and concerns about false positives generate anxiety, discouraging some from participating [29, 37]. Practical barriers, such as inconvenient screening locations, time constraints, and cost misconceptions, hinder access to these programs [37]. Exploring these dynamics within the Italian context is essential to address the specific needs and barriers faced by individuals considering smoking cessation and LCS. This study examines how these challenges manifest in Italy, where cultural, social, and healthcare system factors may uniquely shape CUs' perceptions and decision-making processes.

Recently, the Italian government has initiated pilot programs and pragmatic trials to assess the feasibility and acceptability of LDCT screening [23]—the Italian League against Cancer sponsored and funded a mixed-method study investigating the interactions between primary and secondary prevention. The study involved public smoking cessation clinics in Tuscany and Emilia-Romagna [23]. The study results were convexified into the pilot initiatives promoted by the Ministry of Health, Centre for Disease Prevention and Control (CCM).

This study is a multicenter qualitative research project conducted in two Italian regions. It serves as the qualitative component of a larger initiative the Italian League against Cancer promoted, with the quantitative findings recently published elsewhere [23].

This study presents the findings from the qualitative component, which aimed to explore the perceptions of current and former CUs regarding LDCT screening when offered alongside a smoking cessation program. The central research question was: What are the perspectives of individuals eligible for LDCT screening on the interplay between the two types of interventions (screening and smoking cessation program)?

## Methods

### Study design

Aligned with the research question, qualitative research was conducted using open-ended semi-structured interviews and focus group meetings (FGMs). This approach aimed to triangulate the experiences of individuals participating in an Italian RST and a cessation program named ITALUNG [14] and to gather the perceptions (in terms of acceptability and future care possibilities) of people in the screening target population.

### Sampling and recruitment

Through purposive sampling [38], we included: a) current and former CUs who participated in the ITALUNG study, and b) CUs aged 50–70 with a smoking history of ≥15 pack-years who were offered smoking cessation interventions and were theoretically eligible for future screening. Pneumologists and nurses with expertise in smoking cessation conducted the program in the ITALUNG study (from 2004–2006 to 2008–2010). It included an initial assessment that covered medical history, smoking habits, questionnaires for nicotine dependence, motivation, and self-efficacy, and the Hospital Anxiety and Depression Scale. Participants received smoking cessation counseling and pharmacotherapy, including varenicline, bupropion, or nicotine replacement therapy, over 0–3 months. Six visits were scheduled in the first three months, with further follow-up visits at 6 and 12 months to monitor progress and provide additional counseling [39].

For the interviews, we aimed to purposefully recruit up to 15 trial participants with firsthand experience of the phenomenon under investigation. The target of 15 participants was deemed sufficient to capture a diversity of perspectives from trial participants [40, 41] even for practical reasons: researchers knew that many individuals were unavailable for inclusion due to health-related reasons, including death. The ITALUNG trial participants were aged 55 to 69 years when the trial was approved in 2003, and many had already reached older ages by the time this study was conducted. Despite these limitations, we managed to involve participants who had experienced false positive results during the trial, allowing us to explore a range of perceptions, including those shaped by more challenging experiences in the context of LCS. Investigators from the ITALUNG centers in Florence and Pisa preliminarily contacted participants by telephone. These participants had previously consented to be re-contacted. The investigator explained the research's design and purpose and asked for their willingness to participate. Upon agreement and understanding, the interviewer scheduled an appointment for the participant at a convenient time and location, even online or by phone.

For the FGMs, CUs discussed the LDCT screening proposal alongside the smoking cessation program. Recruitment for the FGMs followed these activities: the local smoking cessation program provider (the Italian League against Cancer—LILT) first reached out to eligible participants (aged 55–70 with a smoking history of ≥15 pack-years, as foreseen for being proposed the LCS in the future), including those enrolled in the program and those who had declined, via phone to explain the research and assess their willingness to participate. This smoking cessation program was based on a cognitive-behavioral approach, beginning with an individual consultation followed by group sessions of 15–20 participants. Each group was preceded by an individual interview to assess smoking habits, dependency, and motivation. The program was structured over two months: the first month included bi-weekly meetings, and the second month had weekly sessions. Afterward, follow-up appointments were scheduled at 6 and 12 months. During the initial phase, participants tracked their smoking habits and identified triggers, using a diary to record their behavior. Throughout the program, participants

were provided with various tools, such as diaries to track their smoking habits, decision-making scales to evaluate the pros and cons of smoking, and information on withdrawal symptoms and nicotine replacement therapies. The program encouraged continued collaboration with general practitioners or specialists for ongoing support in maintaining cessation.

We planned to conduct two FGMs with 6 to 12 [42, 43] participants. The moderator arranged the meeting for those who agreed to participate. During the session, the moderator presented an overview of the study's purpose and provided participants with the information sheet, informed consent form, and privacy form.

## Data collection

Semi-structured interviews comprised a series of open-ended questions for discussing the following:

- Experience of participating in the trial;

- Perception of smoking- and screening-related risks;

- Experience of the diagnostic procedures in terms of teachable moments;

- Perception of the impact of screening;

- Personal needs in terms of support and information for smoking cessation;

- Judgment on the "ethical" dimension of screening.

As to FGMs: before their conduction, we provided the participants with an information booklet from the decision aid developed by the U.S. Agency for Healthcare Research and Quality (https://effectivehealthcare.ahrq.gov/decision-aids/lung-cancer-screening/patient.html), translated into Italian. We started the FGMs by explaining in lay language what LCS with LDCT consists of and what the desired and undesired effects are in the current state of knowledge. Then, the following topics were addressed to moderate the discussion:

- Screening procedures' perceived risks;

- Perception of screening as a teachable moment;

- Personal needs in terms of support and information for prevention;

- Views on smoking cessation programs;

- Opinions on the "ethical" and social dimension of screening.

Research team members who had received specific training conducted interviews and FGMs. They also noted information related to aspects of the person's clinical condition, non-verbal communication, the manifestation of emotions, and contextual events significant to the investigation (field notes). This type of data collection from multiple sources, also known as triangulation [44, 45], allowed researchers to obtain accurate results in both the breadth and depth of information provided by participants.

CB and SM conducted the ITALUNG interviews between March and July 2021, while LG and MEDC moderated the FGMs in June 2021. No physicians or other personnel involved in the participant's care and management participated in the data collection.

## Data analysis

Both interviews and FGMs were audio recorded and immediately verbatim transcribed and anonymized. The data were analyzed using the thematic analysis method [46]. Data analysis

was conducted following a data-driven inductive approach, i.e., reflective thematic analysis (RTA). RTA was adopted because of its flexibility and potential to provide rich and complex understandings [46–48]. An interpretive approach guided the analysis, exploring the social and contextual mechanisms that can inform the construction of meaning systems [46, 48]. Themes were identified at the semantic and latent levels, recognizing concepts directly communicated by participants and considering possible deeper, latent concepts.

Six analytical steps were followed: familiarization with data, initial label generation, theme generation, review of potential themes, theme definition and naming, and report production. LG and MEDC took steps to familiarize themselves with all transcripts, facilitating a comprehensive understanding of the data [49]. MEDC engaged in an extensive review of the entire dataset. Supplementary insight was gained from field notes, aiding in the initial labeling process. Free labeling was employed to capture facets of the data, such as participants' ideas, beliefs, and values, pertinent to the research question. Subsequently, once all relevant data points were labeled, LG generated the initial interpretation of amalgamated meanings, which were discussed with the team members.

Labels were affixed to categorize specific elements across the data, arranging them into provisional themes. The evaluation considered the quality and pertinence of these labels to the dataset and research question. An initial list of themes was compiled to ensure alignment with research objectives, and these were deliberated upon with the other authors. The fifth data analysis stage encompassed mapping and interpretation, identifying meaningful patterns, and discerning central themes. LG articulated theme definitions and crafted a thematic map that depicted the interconnectedness of themes, collectively attributing significance to the phenomenon under scrutiny. In this phase, data excerpts were meticulously chosen by the MEDC and featured in the final report, which was shared and discussed with the research team.

## Rigor

The concept of validity in qualitative research is applied through intersubjective corroboration [50]. All interviewers/moderators received specific methodological training in the development and conduct of qualitative interviewing regarding the relational approach and the most appropriate communication techniques to explore content about the subjective sphere of the participants. The interviews were conducted by female psychologists who managed smoking cessation programs but were unfamiliar with the interviewees. The moderators were qualitative methodologists without prior knowledge of the participants or the interventions, ensuring minimal assumptions and biases. The progress of the interviews and focus groups conducted from time to time were discussed in supervision meetings, focusing on the extent to which the interviewer established and maintained a climate of trust with the interviewee. At least two researchers were involved in each data analysis stage to ensure that the research findings were not merely private interpretations but meant by multiple people and according to various perspectives [45, 51].

## Ethical considerations

The research (in-house prot. n. 2020/0013503 of 03/02/2020) was approved by the Ethics Committee of AVEN (Area Vasta Emilia Nord). This study followed the ICH E6 guidelines for GCP, the principles of the Declaration of Helsinki, Italian and European laws, and reference standards for conducting clinical trials, and it complied with current Italian regulations.

Participants were asked for written consent to participate in the study, including permission to audio record the interview and FGM and take written notes simultaneously. They were assured they could discontinue participation without consequences. Only authorized

researchers viewed, managed, and analyzed the data. Participants were identified with alphanumeric codes, and interviews/FGM recordings were stored anonymously with an identification code.

## Results

### Participants

12 people (5 women, 7 men, > 71 y.o.–mean age: 73, mode age: 75–all retired from work) who had participated in the ITALUNG study with adverse cancer outcomes were interviewed, among whom four had experienced a "false positive" outcome (Table 1). On average, the interviews lasted approximately 27 minutes (ranging from 18 to 52 minutes). 13 people (8 women, 5 men, > 55 y.o.–mean age: 62, mode age: 61) recruited from the cessation program list engaged in two FGMs (Table 2). 6 participants who were not part of any program volunteered to take part, while the others were already enrolled in the group program when the FGMs took place. Both FGMs lasted 2 hours. 15 participants were still CUs at the time of the study.

### Findings

The data analysis yielded six themes covering the views of both groups (interviewed ITALUNG participants and those in the FGMs) on the interactions between the two types of interventions (screening and smoking cessation program). Across their data, we generated the following themes: (i) depreciation and fatalism toward the risk of smoking, (ii) self-blaming and ethicality, (iii) ambivalent impact of the screening on smoking, (iv) LCS-related information and concerns, (v) teachable and motivating moments, (vi) non-stigmatizing communication and testimony by professionals. Table 3 shows the themes with illustrating quotes from the participants.

**Depreciation and fatalism toward the risk of smoking.** Exploring smoking cessation within the context of the LCS proposal highlighted certain attitudes linked to participants' behaviors and their strategies for coping with the prospect of a potential tumor. Participants commonly connected smoking with an elevated risk of cancer, signifying a widespread awareness of the correlation between smoking and lung cancer.

**Table 1. Interviewed participants' characteristics.**

| Code | Gender | Age | Education | Smoking habit | | False positive | Interview modality |
|------|--------|-----|-----------|---------------|-------|----------------|--------------------|
| | | | | At ITALUNG | After | | |
| 01 | F | 75 | Secondary | Yes | No | No | Online |
| 02 | M | 72 | Secondary | Former CU[1] | Yes | No | Online |
| 03 | M | 75 | Tertiary | Former CU | No | Yes | Phone |
| 04 | F | 75 | Secondary | Former CU | No | Yes | Phone |
| 05 | M | 75 | Tertiary | Former CU | Yes | No | Online |
| 06 | F | 73 | Elementary | Yes | Yes | No | Phone |
| 07 | M | 71 | Secondary | Yes | Yes | Yes | Online |
| 08 | F | 74 | Secondary | Yes | Yes | Yes | Face to face |
| 09 | M | 75 | Secondary | Former CU | Yes | No | Phone |
| 10 | M | 72 | Secondary | Yes | No | No | Phone |
| 11 | F | 71 | Secondary | Yes | No | No | Phone |
| 12 | M | 76 | Secondary | Yes | Yes | No | Face to face |

[1] CU = cigarette user

**Table 2. Characteristics of the participants in the focus group meetings.**

| FGM n. | Modality | Gender | Age | Education | Work | Cessation program | Smoking habit |
|---|---|---|---|---|---|---|---|
| 01 | Remote | F | 61 | Secondary | Retired | No | Former CU[1] |
| | | F | 58 | Secondary | Employee | Yes | Yes |
| | | F | 55 | Tertiary | Employee | Yes | Yes |
| | | F | 66 | Secondary | Employee | Yes | Former CU |
| | | F | 62 | Secondary | Housekeeper | Yes | Yes |
| | | F | 64 | Secondary | Retired | Yes | Former CU |
| | | M | 69 | Secondary | Retired | Yes | Former CU |
| 02 | Presence | M | 57 | Secondary | Corporate technician | No | Yes |
| | | F | 59 | Secondary | Store owner | No | Yes |
| | | M | 70 | Secondary | Retired | No | Yes |
| | | M | 65 | Secondary | Self-employee | No | Yes |
| | | M | 61 | Secondary | Employee | Yes | Former CU |
| | | F | 64 | Tertiary | Retired | No | Yes |

[1] CU = cigarette user

"It brings cancer to my mind; I immediately associate it with that; then I know that maybe it's not the only problem smoking causes because there can be many issues, but cancer is the scariest thing, a bit of a bogeyman, perhaps the most well-known thing"

(FG01).

Participants acknowledged a level of worry or apprehension about the risk of a tumor, but their concern did not reach an extreme level of distress. Participants recognized the potential dangers of smoking and lung cancer, yet their concern was not overwhelming or paralyzing.

"Risk eh... But to tell the truth, I haven't thought much about it, I was thinking about quitting, that's it"

(07).

The participants showcased a spectrum of attitudes concerning risk, ranging from trivializing the hazards of smoking to embracing a sense of inevitability about the outcomes.

A subset of participants (CUs who did not welcome the smoking cessation program) conveyed a feeling of surrender regarding the detrimental impacts of smoking, potentially indicating their willingness to acknowledge and come to terms with the adverse consequences.

"It is not true that the lung comes back the way it was"

(08).

"I think that a smoker of a certain age, after forty or fifty years or even more than fifty years of smoking, is aware of the risk, but thinks about it occasionally, not always"

(FG02).

**Self-blaming and ethicality.** Regarding the views on the screening proposition, participants offered observations on its overall acceptability, drawing parallels with other screening initiatives they have typically embraced previously.

**Table 3. Themes and illustrative quotes.**

| Theme | Quote(s) |
|---|---|
| Depreciation and fatalism toward the risk of smoking | "Always calm, (*laughs*) anyway, sooner or later we have to die (*laughs*)" (01)<br>"You always think it happens to others and not to you. Because if you think about it—and I knew it was harming me—but it's like feeling immortal, like, 'it won't happen to me.' But that's not true because you, in turn, are part of the 'others.' So, that was my perception." (FG02) |
| Self-blaming and ethicality | "As a smoker, I decide to smoke, and there's continuity, so it's not like sunbathing; maybe you could get a sunburn, but you don't do it every day. Cigarettes are different [. . .] you know you're making a mistake. The idea of undergoing an examination that can make you understand or confront an ugly reality is scary. You realize that the blame is solely yours. You can't say it's the sun's fault or. . . it's my fault! And this scares you!" (FG02)<br>"But dying from lung cancer, that really scares me, and I also live with a sense of guilt, in the sense that dying from cancer and leaving my three children, even though they are grown, I would still remain the only pillar, you know." (FG01)<br>"I would distinguish for a moment: does the screening serve to prevent deaths, or does the screening serve to discourage smoking? In the first case, I think it's useful, then it's true, there are false positives, etc., but like all mass screenings, I think it's useful to try to catch a problem at the beginning and prevent deaths. So, personally, I'm in favor" (FG01) |
| Ambivalent impact of the screening on smoking | "Then, when I reached mental and physical maturity, I started having check-ups. I would practically have chest X-rays, and since the results were always good, I also kept getting my throat checked. As long as the results were good, I used to say: why should I quit smoking?" (FG01)<br>"For those who are addicted to smoking, they definitely have this thought: they say, 'Great, everything turned out fine, I can keep smoking'" (FG01) |
| LCS-related information and concerns | "As A. used to say, when smokers are scared, the first thing they do is smoke a cigarette. So, if you scare me like that right away, it's probably not easy to handle emotionally. It always depends on how it's presented, I think" (FG02)<br>"But perhaps at this point, it would be even more frightening not to know that you have an illness and end up being one of those three (positive). Maybe it also depends on how one sees it" (FG02)<br>"Actually, I was also a bit stuck on the data because just 3! I thought it would be a bit higher! Honestly, those aren't very encouraging numbers!" (FG01) |
| Teachable and motivating moments | "I believe, however, that over time, as medicine continues to improve, screening becomes increasingly important. I also think it might, paradoxically, help even more those people who want to quit smoking but struggle to do so. In the sense that it's something you experience personally, so it could help—more than just seeing images—because it's something you're actively doing. It gives you a greater awareness of what you're doing." (FG01) |
| Non-stigmatizing communication and testimony by professionals | "Now this thing comes up. I'm glad, but before we were really discriminated against by this hospital. . . There was even a sign outside, in the radiology department, in the new section, saying: 'Chest CT scans are not performed on smokers.' Not even for a fee." (FG02) |

"To be honest, I was feeling well. As for the screenings, I agreed since I did them all and accepted this thing. It was a good thing; they did the CT scan, the spirometry, and all these things"

(05).

"I think that I would like to do the screening because I would try to see if it gives me any problems if I have any problems, just like I do the colon one like we do the others"

(FG01).

However, for the participants of FGMs, this screening was unique in the dynamic it triggered, which was characterized by self-blaming.

"Other screenings are for diseases you didn't inflict on yourself, let's say. While with smoking, you know that if, unfortunately, something obvious were to come up, you'd also have a sense of guilt, the weight of knowing that you brought it upon yourself. So, there's this difference. Personally, if I had to do a screening for other things, I myself would have less problems doing it"

(FG01).

The perceived self-blame and the intricate emotional response to possible adverse results underscored the complex decision-making dynamics. Given this self-blaming, FGMs' participants articulated some ethical considerations regarding the LCS proposal. Their distinction between the purpose of screening—either preventing deaths or discouraging smoking— highlighted the moral complexity of this public health intervention. All participants stated that proposing screening to CUs was justified.

"I think that the money isn't wasted"

(10).

"Because it's the right thing to do, especially for those who have been smoking for many years and never get checked, it's right to have a screening, even for the lungs"

(04).

**Ambivalent impact of the screening on smoking.** According to our participants, the screening had an ambivalent impact on smoking behaviors and decisions. When considering smoking cessation in the lung cancer screening proposal, participants exhibited a range of reactions and attitudes. On the one hand, the proposal temporarily heightened awareness of the need to quit smoking. Participants recognized that the screening process could motivate them to stop due to the potential implications it revealed.

"Well, I considered it a positive thing because gradually it encourages you to quit smoking, to realize the seriousness of smoking, which is now known everywhere. A few years ago, there was less awareness, even a few years ago. . . but there was less. . . You could still smoke anywhere. Gradually, the screening makes you think about quitting smoking"

(02).

This heightened consciousness wasn't consistently sustained, as a subset of participants noted that the fear eventually waned, leading to a reduced emphasis on potential health hazards, thereby limiting the motivating influence of the screening process for smoking cessation.

"But the fear passes, and then you don't think about it much. If you must think about what could happen when you wake up, you just stay in bed"

(10).

In this context, the proposal further strengthened a decision-making process that was already underway.

"If they didn't find anything, you might let out a sigh of relief and say maybe it's time to quit; so far, I've been lucky"

(FG02).

On the other hand, several participants concurred that using health monitoring as a justification to continue or resume smoking was evident, mirroring the sentiments of numerous interviewees.

"They told me 'You have to quit!'. Yes, they're right, I must quit, and then you don't quit, and you smoke. But since my lungs were always clear, I kept going"

(12).

**LCS-related information and concerns.** There was a distinction between individuals involved in ITALUNG, who had experience with LDCT, and those consulted in the FGMs, particularly concerning the necessity for clarity on specific aspects of LCS. They requested additional clarification on false positives and the percentage-based mechanism underlying the screening. Notably, their concerns centered around potential false positive outcomes, the prospect of undergoing biopsies, and comprehending the significance of the screening rates.

"I am concerned about the three hundred and fifty false positives, in the sense that maybe if I could choose to push myself to undergo an examination, I would prefer more definite certainties than these. Because if I must endure a hell of a scare because I'm shown to have a tumor when I don't"

(FG01).

The participants expressed concern about the high rate of false positives in the screening process, highlighting the uncertainty and fear of potentially receiving a false cancer diagnosis. They emphasized the psychological impact of such results and the desire for a more reliable testing method.

"300 out of 1000 is a lot! If there's a possibility of improving the technology, these false positives may decrease. These are false positives where the psychological impact is quite significant on the people who receive this type of diagnosis, and then in the end, it turns out they weren't risking their lives"

(FG02).

**Teachable and motivating moments.** The smoking cessation program was relatively less discussed among the participants. The participants in the FGMs who attended a program found the smoking cessation program meetings motivating experiences. In contrast, those who dropped out and those invited to participate in the trial and attended the program showed a degree of criticism as they felt it wasn't the optimal time for them to do so. The ITALUNG participants expressed a positive sentiment toward the guidance provided by their general practitioners and the significance of follow-up appointments. These interactions have

contributed to their increased awareness of their smoking behavior and motivated them to align their actions with the advice given during previous consultations.

> "The CT scans didn't influence. . . the consultations with my doctor had a greater impact. Also, because the doctors were brainwashing a bit (laughs), and so on. . . The fact that I had to come back for a follow-up and tell them if I had decreased [smoking] made me aware of what I was doing. So, I tried to present myself at the appointments based on what they had told me the previous time"

> (11).

Beyond the screening proposal's already mentioned role, interviewees perceived these interactions as teachable moments, providing insights and support for their efforts to quit smoking or maintain their non-smoking status. Regular medical consultations were seen as opportunities for education and motivation in the context of smoking cessation. Both participating cohorts expressed favorable opinions about proposing the LCS combined with a smoking cessation program. Still, some in the FGMs emphasized keeping program attendance non-mandatory.

**Non-stigmatizing communication and testimony by professionals.** Connected to the previous theme, participants' desire to turn their interactions with healthcare providers into teaching opportunities and foster motivation is evident in their requests for specific qualities in these interactions. Their preference for lung cancer screening to be conveyed in a manner that goes beyond solely targeting CUs and instead emphasizes the importance of lung health for everyone reflects a desire for communication that avoids stigmatization. Their discontent with previous discriminatory practices has spurred their advocacy for a more inclusive approach to promoting lung health.

> "Then there was, I don't remember the exact timing, but the last visit I had was about three years ago, well, they even had me see a pulmonologist who did a CT scan and advised me to quit, and that there were risks, he said, 'You're asking for it' or something like that, 'but we don't do checks anymore, you've turned 70'"

> (07).

> "In my opinion, it should be presented as prevention not just for smokers, but because the lungs are an organ and need protection, but not focusing solely on smokers because otherwise, it's like always beating up on people who smoke. I think it should be approached a bit differently, just like prevention for all other organs"

> (FG01).

In this context, participants shared their perspectives regarding the involvement of doctors and general practitioners. A subset of participants conveyed their dissatisfaction with healthcare professionals who, despite being CUs themselves, offered advice on smoking cessation. Conversely, some participants underscored the significance of physicians setting an example through their behavior. Additionally, a few participants recounted personal stories highlighting the juxtaposition between medical counsel and actual behavior.

> "Doctor S., he smoked and told that patient "you have to quit smoking." But come on, I mean, don't smoke in front of me!"

> (FG02).

"All the doctors, including my GP who enrolled me in this study, he smoked 40 cigarettes a day and told me, 'You should do what the doctor says, not what the doctor does'"

(04).

## Discussion

This research explored the perspectives of participants eligible for LCS and a smoking cessation program regarding their combined proposal. The findings of this study contribute to the broader field of smoking-related lung cancer stigma by confirming and extending existing knowledge, particularly regarding the role of stigma and self-blame in individuals' perceptions of smoking, lung cancer, and the role of the screenings [52–56].

A particularly novel finding from the Italian context is the prevalence of fatalism. This finding highlights a spectrum of attitudes towards smoking and lung cancer risk, ranging from downplaying the dangers to accepting them as unavoidable. While participants acknowledged the risks associated with smoking, their emotional responses and coping mechanisms varied, with some displaying resignation or a lack of urgency to quit, indicating a complex relationship with both the prospect of lung cancer and smoking cessation efforts. In Italy, we observed a prevailing sense of fatalism among participants, contrasting with the more individualistic and proactive attitudes often seen in Anglo-Saxon countries [57, 58], even not absent [59]. This fatalism, rooted in historical and cultural perspectives on health and disease [60–62], contributed to a complex emotional response to smoking cessation and LCS. Fatalism is seen as a coping style characterized by low levels of anxiety and depression, low sense of control, resignation, and passive acceptance of fate [63, 64]. Research indicates that individuals with a heightened fear of cancer and fatalistic attitudes are less likely to engage with information about advancements in cancer control, perpetuating their negative emotions and perspectives [65]. Moreover, fatalistic beliefs may be associated with a higher risk of advanced-stage LC diagnosis [66].

Participants often expressed ambivalence toward the potential impact of screening on smoking behavior, which was not merely the result of self-blame but was deeply intertwined with culturally ingrained notions of fate and control over one's health. This finding is notable, as it highlights how cultural factors such as fatalism can shape perceptions of lung cancer risk and the willingness to engage with screening, ultimately influencing the effectiveness of public health interventions.

In addressing the broader field of smoking-related lung cancer stigma, this study makes a significant contribution by expanding our understanding of the complex dynamics between smoking, LCS, and stigma [27, 29, 34, 37, 58, 59], particularly within the Italian context. The widespread self-blame observed in this study may be partly a result of the emotional dynamics triggered by individuals' interactions with healthcare services [67]. Many participants linked their smoking behavior to feelings of guilt and shame, a pattern consistent with findings from other studies where healthcare encounters serve as significant emotional triggers [68]. This suggests that when individuals engage with healthcare providers, especially in the context of screening programs, their awareness of risky behaviors like smoking may lead to a heightened sense of culpability [67, 69, 70]. Screening programs, which often reveal hidden or neglected health risks, are not neutral interventions for people who are already conscious that their actions may harm their health. Instead, these screenings can serve as poignant reminders of personal responsibility, sometimes intensifying emotional responses like guilt or regret, especially when the outcomes are perceived as a consequence of their behaviors [71, 72]. The participants in the FGMs, unlike those in the ITALUNG trial who had prior experience with LCS, perceived the screening differently, primarily due to the self-blame it triggered, contrasting it

with screenings for other diseases, as also shown by a survey by Raz and colleagues with CUs in California [35]. Our participants in the FGMs expressed feelings of guilt, acknowledging smoking as a self-inflicted risk factor, which heightened their concerns about the screening's potential outcomes.

In the context of this study, self-blame emerges as a prospective and undesirable outcome that individuals eligible for screening aim to avoid [69]. Self-blame, a form of causal attribution, arises when individuals perceive personal control over the cause of an event, often occurring in chronic health conditions [73]. It entails an individual believing that an unwanted event is somehow their fault, indicating personal responsibility for its occurrence, as shown by LC patients involved in a study by Lehto [55]. This sequential decision-making process involving perceived control and responsibility applies when individuals form self-blame perceptions following a health diagnosis, such as cancer [74].

Furthermore, our study presents unique insights into the role of healthcare professionals in shaping perceptions of LCS. Participants in our study emphasized the importance of non-stigmatizing communication and empathetic interactions with healthcare providers, echoing findings in other contexts but also highlighting the necessity for such approaches in a society where smoking-related diseases are often stigmatized [70]. This cultural specificity is essential for understanding how stigma is sustained and addressed in Italy, yet it remains largely overlooked globally [67–69].

These understandings underscore the importance of avoiding situations that may lead to self-blame. Healthcare professionals must recognize that self-blame can serve as a potential barrier, particularly in health interventions such as LCS [75] and smoking cessation programs [76]. Awareness of this possibility allows professionals to mitigate feelings of self-blame among eligible participants proactively [35]. By providing clear and supportive communication, addressing misconceptions, and emphasizing the multifactorial nature of health conditions, healthcare professionals can help prevent individuals from experiencing unnecessary guilt or self-blame.

Regarding the impact of LCS on smoking behavior, our participants exhibited a diverse array of responses. While some initially felt motivated to quit smoking due to heightened awareness, others experienced a gradual decline in fear, resulting in diminished concern about health risks. Notably, 7 out of 12 ITALUNG participants who initially expressed motivation to quit ultimately continued or resumed smoking, citing the absence of immediate health issues as justification. FGM participants echoed similar concerns and contemplated LCS as a potential future scenario. This observation resonates with findings from previous studies, indicating how a negative LDCT screen can instill a false sense of security regarding an individual's risk of developing lung cancer [27, 34, 37, 77].

As to LCS per se, individuals engaged in the FGMs conveyed a requirement for elucidation regarding several facets, especially false positives and the effectiveness of the screening process. Their apprehensions revolved around the possibility of false positive results, the likelihood of undergoing biopsies, grasping the importance of screening rates, indicating a preference for more dependable testing approaches, and minimizing psychological repercussions. Our findings suggest how proposing LCS necessitates a transparent and comprehensible information process, steering clear of reducing the screening moment to a mere checkbox activity, as anticipated by previous studies [19, 39, 76, 78]. Instead, it should lay the groundwork for a meaningful teachable moment [79]. To achieve this goal, comprehensive training for healthcare professionals is imperative, ensuring they possess the necessary skills to effectively communicate the significance of LCS to patients.

Additionally, healthcare providers should have the knowledge and resources to facilitate meaningful discussions about smoking cessation during screening [39]. Considerations should

include sensitivity to patients' concerns, addressing potential barriers to quitting smoking, and providing personalized support and guidance tailored to each individual's needs and circumstances [19, 39, 78]. In this regard, the Trans Theoretical Model (TTM), developed by DiClemente and Prochaska [80], has been recently adopted by organizations such as LILT and various public health services. It serves as a beneficial framework to assist individuals in their process of change, particularly in the context of motivation to quit smoking. This model can provide a valuable foundation for conducting brief motivational interviews during various healthcare encounters, including consultations with general practitioners, screenings, and follow-ups.

Moreover, it is essential to ensure that the smoking cessation program is proposed at the right time, as the study's participants reported. The smoking cessation program received less attention during our data collection, with participants in the FGMs who attended finding it motivating while others criticized its timing. As demonstrated elsewhere [76], eligible individuals may not be psychologically prepared to quit smoking, resulting in diminished motivation to accept cessation support. TTM can offer a conceptual paradigm for discerning the most appropriate moment for proposing and providing such interventions [81].

Careful consideration of timing and persons' readiness to quit using cigarettes [80] fosters a supportive and empowering environment for individuals undergoing screening, ultimately enhancing their overall experience and engagement with healthcare services. Our findings indicated that moments of engagement with healthcare professionals were vital in motivating participants. Specifically, ITALUNG participants valued guidance from general practitioners and follow-up appointments, which heightened their awareness of smoking behavior and provided motivation. Interaction with general practitioners, attendance at smoking cessation programs, and medical consultations during the trial were opportunities to learn and make positive health decisions [39].

Finally, participants' desire to turn interactions with healthcare providers into teachable moments, fostering motivation, is evident in their requests for specific qualities in these encounters. They prefer that LCS communication emphasizes lung health for everyone, aiming to avoid stigmatization. Discontent with past discriminatory practices has fueled advocacy for a more inclusive approach to promoting lung health. Participants express perspectives on the involvement of doctors and general practitioners, expressing dissatisfaction with healthcare professionals who smoke yet offer advice on smoking cessation. In contrast, others stress that physicians set an example through their behavior.

This finding underscores the significance of the quality of relationships with healthcare professionals, whether general practitioners or specialists, in facilitating effective communication and support for LCS-eligible individuals. It emphasizes the need for healthcare professionals to align their behaviors with the advice they give, thereby enhancing credibility and trust among patients. Additionally, it highlights the importance of healthcare professionals being sensitive to persons' experiences and preferences, fostering a supportive and non-judgmental environment conducive to behavior change. Therefore, training for healthcare professionals in effective communication strategies and awareness of the impact of personal behaviors is essential for promoting positive health outcomes and patient empowerment.

## Strengths and limitations

This study benefitted from a comprehensive data collection approach, including individual interviews and FGMs, allowing for a nuanced exploration of different participants' perspectives. Notable limitations of the study include the general self-selecting bias for both the studies and the potential recall bias among participants in the ITALUNG trial. Self-selection bias

occurs when individuals who choose to participate in a study are not fully representative of the broader population, as their decision to join is often driven by personal motivations or interests. In the case of these studies, this bias is evident in the participants of the ITALUNG trial and the FGMs members, as they were likely motivated by factors such as an interest in smoking cessation or LCS. While this bias may restrict the generalizability of the findings, it is a typical aspect of qualitative research, which prioritizes deep insight into specific experiences or viewpoints rather than statistical representativeness. Additionally, some participants who opted not to engage in smoking cessation programs provided an underrepresented perspective, offering valuable insights that are often overlooked in other studies. Regarding recall bias, since participants of the ITALUNG trial were asked to reflect on their screening experiences retrospectively, their recollections may have been influenced by time and individual perceptions. Additionally, the study's findings may not be generalizable to populations that do not overlap with the participants' characteristics and share similar backgrounds and life histories. Finally, as with any qualitative study, there is the possibility of researcher bias influencing the interpretation of the data, although efforts were made to mitigate this through intersubjective corroboration and transparency in the analytical process.

These findings provide comprehensive insights into participants' perspectives on LCS and smoking cessation programs, covering concerns, attitudes toward risk, the impact of the screening proposal, and interactions with health professionals. The study contributes to the international debate on LCS programs by offering a nuanced understanding of these complex factors, particularly within the context of Italy. Exploring these aspects is vital to crafting effective strategies for promoting lung cancer prevention, early detection, and effective smoking cessation programs.

This study underscores the importance of refraining from stigmatizing CUs and honoring their dignity. CUs should not be solely held responsible for smoking; instead, they should recognize that the healthcare system is committed to their overall health, particularly their respiratory well-being, irrespective of any perceived culpability. Implementing this approach would enhance the acceptability of interventions targeting CUs and foster a supportive and empathetic environment, ultimately leading to more favorable health outcomes.

## Supporting information

**S1 Checklist. COREQ checklist.**
(DOCX)

## Acknowledgments

Working Group members and collaborators:

Group members: Eugenio Paci, Sandra Bosi, Paolo Giorgi Rossi, Luca Ghirotto, Alessandro Peirano, Ermanno Rondini, Francesco Rivelli, Fabio Falcini, Antonio Nicolaci, Angela Zannini, Gerardo Astorino, Marco Tamelli, Salvatore Cardellicchio, Laura Carrozzi, Francesco Pistelli, Patrizia Gai, Giacomo Lavacchini.

Group collaborators: Matteo Ameglio, Cristiano Chiamulera, Giovanni Forza, Francesco Torino, Giuseppe Gorini, Donella Puliti, Giovanna Cordoprati, Olivera Djuric, Elena Camelia Ivanciu, Matías Eduardo Díaz Crescitelli, Claudia Bricci, Silvia Marini, Simonetta Salvini, Elisabetta Bernardini, Chiara Cresci, Valentina Bessi, Valentina Galli, Silvia Stoppa, Francesca Zironi, Andrea Lopes Pegna.

## Author Contributions

**Conceptualization:** Luca Ghirotto, Eugenio Paci, Francesco Pistelli, Giuseppe Gorini, Sandra Bosi, Paolo Giorgi Rossi.

**Data curation:** Luca Ghirotto, Claudia Bricci, Silvia Marini, Valentina Bessi, Matías Eduardo Díaz Crescitelli, Francesco Pistelli, Paolo Giorgi Rossi.

**Formal analysis:** Claudia Bricci, Silvia Marini, Valentina Bessi, Matías Eduardo Díaz Crescitelli.

**Funding acquisition:** Eugenio Paci, Ermanno Rondini, Giuseppe Gorini, Sandra Bosi, Paolo Giorgi Rossi.

**Investigation:** Luca Ghirotto, Eugenio Paci, Claudia Bricci, Silvia Marini, Valentina Bessi, Matías Eduardo Díaz Crescitelli, Giuseppe Gorini, Paolo Giorgi Rossi.

**Methodology:** Luca Ghirotto, Matías Eduardo Díaz Crescitelli, Ermanno Rondini, Francesco Pistelli, Paolo Giorgi Rossi.

**Project administration:** Luca Ghirotto, Eugenio Paci, Valentina Bessi, Matías Eduardo Díaz Crescitelli, Ermanno Rondini, Sandra Bosi, Paolo Giorgi Rossi.

**Resources:** Claudia Bricci, Ermanno Rondini, Francesco Pistelli, Giuseppe Gorini, Sandra Bosi, Paolo Giorgi Rossi.

**Supervision:** Luca Ghirotto, Eugenio Paci, Valentina Bessi, Giuseppe Gorini, Sandra Bosi, Paolo Giorgi Rossi.

**Validation:** Eugenio Paci, Silvia Marini, Valentina Bessi, Ermanno Rondini, Paolo Giorgi Rossi.

**Visualization:** Luca Ghirotto, Matías Eduardo Díaz Crescitelli.

**Writing – original draft:** Luca Ghirotto, Claudia Bricci, Silvia Marini, Valentina Bessi, Matías Eduardo Díaz Crescitelli, Ermanno Rondini, Francesco Pistelli, Giuseppe Gorini, Sandra Bosi, Paolo Giorgi Rossi.

**Writing – review & editing:** Eugenio Paci, Paolo Giorgi Rossi.

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
