## [Decision Letter · Decision Letter 0]

28 Nov 2024

PONE-D-24-38765Self-blaming as a Barrier to Lung Cancer Screening and Smoking Cessation Programs in Italy: A Qualitative Study

PLOS ONE

Dear Dr. Ghirotto,

Thank you for submitting your manuscript to PLOS ONE. After careful consideration, we feel that it has merit but does not fully meet PLOS ONE’s publication criteria as it currently stands. Therefore, we invite you to submit a revised version of the manuscript that addresses the points raised during the review process.

I have examined the paper myself and was able to secure the services of 2 expert reviewers. As you can see, they largely agree on the strengths of the paper and I concur. Reviewer 2 has provided an extensive list of useful suggestions, including a set of points that need to be referenced with which I strongly concur (e.g., Globocan will provide the most current incidence and prevalence figures for Italy etc). My suggestions are as follows:

Please address each of the reviewers’ points in turn. I will add some further comments on some of these below, as well as comments of my own.The abstract is definitely too long. The limit for PlosOne is 300 words and you are at about 430. This needs to be trimmed and certainly can be trimmed.Please go through and address the paragraph structure of the manuscript. Avoid any single sentence paragraphs. These must be integrated with the paragraphs below or above, as appropriate, or inserted elsewhere.As you can probably tell by my name, I too can see the mixed English/Italian idiom. I strongly suggest you remove this entirely. In general, second language (L2) speakers have difficulty in comprehending idioms and so these should be avoided altogether in scientific and scholarly papers. That is, I would also suggest you remove “the straw the broke the camel’s back”.The sample size needs to be introduced carefully because it also points back to a process that is very important in qualitative research that has not been addressed in your manuscript, that of saturation. The reviewers ask why you chose 15 people. In general, in interview studies such as yours, the researchers would commence gathering interview data and only stop once each it was evident that no new material was forthcoming. This is called the point of saturation. It could occur after 5 or 25 or more interviews. It is difficult to reconcile how you would specify 15 people before you began interviewing and then have a sense of saturation. It is OK if you needed to, for pragmatic reasons, limit the number of interviews. But you need to explain that clearly and also to discuss what approaches you took to assess saturation in the circumstances.For the TA, you should present a table with all the themes and an illustrative quote for each. This is in addition to the material in the body and will greatly help readers navigate your model.Further, I think that the Discussion section misses the opportunity to relate cultural factors to psychological perceptions of screening, cancer, and lung cancer. In my current position in Hong Kong, I have been exposed to some distinctly Chinese perspectives on cancer that has contrasted with results of my previous work with lung cancer patients in Australia. Similarly, you might consider what aspects of the Italian sample and their interpretations can be drawn out. Much of the scholarship in this area is dominated by the Anglophone world and it is important to give voice to true cultural factors. For example, the potential for Italian fatalism seemed to be evident in some of the quotes I saw.As for the references to the Youtube video and other public health activities based on this study, these should be removed from the manuscript. They are certainly laudable, but it is up to the organization to promote these and to derive impact from this research. This does not belong in the Strengths and Limitations section, nor in any other part of the paper.Finally, you need to apply a qualitative assessment framework such as COREQ and incorporate it into the MS, most likely as an Appendix. This is very important for readers to be able to contextualise your work.

We look forward to receiving your revised manuscript.

Kind regards,

Stefano Occhipinti

Academic Editor

PLOS ONE

Journal Requirements:

 This study contributed, without receiving funding, to the Italian Ministry of Health Pilot project “Progetto Pilota di un programma di screening per il tumore polmonare integrato con la cessazione del fumo: percorsi, selezione dei soggetti e protocolli diagnostici, in vista di una valutazione HTA”. The study was partially funded by Emilia Romagna Regional Health Authority DGR n° 1800/2020.

The research project “Analisi dei meccanismi d’azione psico-comportamentali con cui la proposta di adesione alla TAC spirale agisce sulle abitudini tabagiche e sulla disassuefazione al fumo” has been supported by a ‘5x1000’ national grant from the Italian League against Cancer (LILT-Rome).

This study was partially supported by the Italian Ministry of Health – Ricerca Corrente Annual Program 2025.  

This study contributed, without receiving funding, to the Italian Ministry of Health Pilot project

“Progetto Pilota di un programma di screening per il tumore polmonare integrato con la cessazione

del fumo: percorsi, selezione dei soggetti e protocolli diagnostici, in vista di una valutazione HTA”.

The study was partially funded by Emilia Romagna Regional Health Authority DGR n° 1800/2020.

The research project “Analisi dei meccanismi d’azione psico-comportamentali con cui la proposta

di adesione alla TAC spirale agisce sulle abitudini tabagiche e sulla disassuefazione al fumo” has

been supported by a ‘5x1000’ national grant from the Italian League against Cancer (LILT-Rome).

This study was partially supported by the Italian Ministry of Health – Ricerca Corrente Annual

Program 2025.

 This study contributed, without receiving funding, to the Italian Ministry of Health Pilot project “Progetto Pilota di un programma di screening per il tumore polmonare integrato con la cessazione del fumo: percorsi, selezione dei soggetti e protocolli diagnostici, in vista di una valutazione HTA”. The study was partially funded by Emilia Romagna Regional Health Authority DGR n° 1800/2020.

The research project “Analisi dei meccanismi d’azione psico-comportamentali con cui la proposta di adesione alla TAC spirale agisce sulle abitudini tabagiche e sulla disassuefazione al fumo” has been supported by a ‘5x1000’ national grant from the Italian League against Cancer (LILT-Rome).

This study was partially supported by the Italian Ministry of Health – Ricerca Corrente Annual Program 2025.

4. In the online submission form, you indicated that The data that support the findings of this study are available on request from the corresponding author, LG. 

Reviewers' comments:

Reviewer's Responses to Questions

**Comments to the Author**

1. Is the manuscript technically sound, and do the data support the conclusions?

Reviewer #1: Yes

Reviewer #2: Yes

2. Has the statistical analysis been performed appropriately and rigorously? 

Reviewer #1: Yes

Reviewer #2: N/A

3. Have the authors made all data underlying the findings in their manuscript fully available?

Reviewer #1: No

Reviewer #2: No

4. Is the manuscript presented in an intelligible fashion and written in standard English?

Reviewer #1: Yes

Reviewer #2: Yes

5. Review Comments to the Author

Reviewer #1: Overall, the authors present a rigorous qualitative study that explores the perspectives of stakeholders involved in LCS and individuals eligible for the smoking cessation program. They seem to recommend implementing timely smoking cessation programs alongside LCS initiatives, with an emphasis on effective communication methods to help reduce potential self-blame among participants. Please see my feedback for minor revisions below.

Suggested minor revisions:

Introduction

Authors wrote “The results of previously published qualitative studies indicate the need to consider the specific needs of individuals who decide to embark on a smoking cessation journey [16,22–24,29], given the uniqueness of the target population”. Could you elaborate on this statement and give examples of those specific needs and link these findings to the aim and justification of your study?

Methods

Authors wrote “Through purposive sampling [30], we included: a) current and former cigarette users who participated in the ITALUNG study, and b) cigarette users aged 50-70 with a smoking history of ≥15 pack-years who were offered smoking cessation interventions and were theoretically eligible for future screening.” If data is available, please include details on when the smoking cessation interventions were offered to the participants (how many years ago?, how long has it been since the participants have been involved in these programs (ITALUNG, smoking cessation?)

Authors wrote “Semi-structured interviews comprised a series of open-ended questions for discussing the following:

- Experience of participating in the trial.

- Perception of risk.”

Please specify what risk is being referred to here (e.g., participants’ own perception towards their risk towards lung cancer? Or risk of participating in the trial?)

Results

For the tables, please include notes for abbreviations used (e.g., What is Ex?).

Discussion

How do the findings contribute to the broader field of smoking-related lung cancer stigma beyond the existing knowledge on the association between smoking and self-blame? It would also be helpful to highlight any unique findings from your study compared to previous research in this area, such as culturally specific observations or unique insights from conducting this study in Italy compared to studies in other contexts.

Minor typos:

Introduction – 2nd paragraph: After the first “random screening trial” is mentioned, please bracket RST for future abbreviations.

Reviewer #2: This is a solid qualitative study about Italians´ perception of both lung cancer screenings and smoking cessation interventions. I think authors did a great job in conducting the study and presenting their results. The rationale and the need for the research is also well-explored, as the author discussed the possible implementation of lung cancer screening as an intervention program in Italy.

That being said, I only have few minor suggestions that I think could further enhance the overall quality of the paper. Some of these are a bit more substantial, while most are actually just minor typos.

More substantial:

1) It is generally recommended to avoid excessively short paragraphs, especially those with just a single sentence. The authors tend to use many full stops, even when they are still discussing the same topic. I suggest reviewing the text with this in mind.

2) The abstract is very detailed, but quite lengthy. I am not updated about the journal´s policy about the maximum length of the abstract, but I will suggest to revise the result and conclusion sections to shorten them up.

3) At the same time, please mention the analysis technique you employed (RTA) in the abstract.

4) There are some points throughout the manuscript when I expected to find a citation, but none is provided. Following, I will report the sentences that in my opinion should be appropriately addressed to a reliable quote: I strongly suggest authors to add them up.

a. Lung Cancer (LC) is the first cancer cause of death in Europe.

b. Smoking is the most decisive risk factor and has the most considerable attributable fraction in most countries.

c. In recent decades, there has been a noticeable reduction in age-adjusted incidence rates of lung cancer.

d. Early diagnosis has the potential to lower disease fatality rates, although attempts in the 1970s were largely unsuccessful.

e. RTA was 9 adopted because of its flexibility and potential to provide rich and complex understandings

5) Page 7. Authors state they aimed at recruiting 15 participants from the trial, but there is absolutely no rationale mentioned behind this choice. This problem is mirrored with focus groups. Please provided an explanation for your sample size choices within the text, with proper references.

6) Related to prior point, please also add information about how the composition of each focus group was determined and why.

7) At page 13, just before the first sub-paragraph about the emerged themes (Depreciation and fatalism toward the risk of smoking) there is a text paragraph which is meant to anticipate the results (starting with “A prevailing sense of optimism…” until the end). I think this might be bad positioned here, as it basically synthetizes the upcoming results. I would consider deleting it or moving it in the discussion section.

8) Page 11: “Six participants were proposed with the program but did not accept it”. Do you have any information about their reasons? Or their demographics? This should be addressed in the limitation as a self-selecting bias.

9) Page 24, the first paragraph of strength and limitations (from “This qualitative analysis has proven to be a powerful tool for communication…” till the last YouTube link). I am not convinced this is the right place to mention the use of this research findings for dissemination. In my understanding, this is the place where you should discuss about the strength more related to the study design and implication for future research and scientifically-oriented interventions. Maybe you could mention this information – including the YouTube video – in a footnote?

Minor suggestions/typos:

1) Page 5, “Qualitative research delves into participants' narratives to provide insights into those elements, which are invaluable for designing interventions that resonate with patients' real-world experiences”. This sentence sounds odd. It is unclear what “those elements” is referred to. Please rephrase to improve clarity.

2) Page 6, there is a space missing between “Emilia-Romagna” and the citation 15.

3) Page 6, “The central research question was: What are the perspectives of individuals eligible for LDCT screening on the interplay between the two interventions (screening and smoking cessation program)?”. I would suggest to add “kind of” or “type of” before “interventions”. This expression is used also in other places of the manuscripts but in my opinion sounds odd.

4) Page 7, “The investigator explained the research's type and purpose”. I suggest to revise the use of the word “type” here. Maybe “design” or “methodology” would work better.

5) For consistency, please revise the use of periods in the excepts from participants´ words. I suggest avoiding the period at the end of quoted sentences (i.e., within quotation marks), and using it directly at the end after the parentheses indicating the participant information (e.g., FGM01 or 01).

6) Pages 7-8. I would suggest to use semi-column instead of periods in the bulleted lists.

7) Page 17 “the last straw that definitively overflowed the vase”. I think that an international audience would not understand this, as it appears to be a mix between a literal translation of the Italian idiom “la goccia che fa traboccare il vaso” and its respective in English, “the straw that broke the camel's back”. If I am wright, I would suggest author to just choose one of the two and use it in full!

8) Page 8. You mention that the “interviews and FGMs were immediately verbatim transcribed”, but if I am right you have never mentioned before that these were audio and/or video recorded. If this is the case, please add more information about this in the text.

9) ACKNOWLEDGMENTS, first line: “This study contributed, without receiving funding, to..”. I would suggest to rephrase it making clear that no founds have been received specifically from that project, as than other funding are listed.

10) While presenting the sample, the number of males and females in bracket is sometimes written in numbers and in other cases in letters. Please be consistent (I would choose numbers, but I am not sure about the journal policies if there are any).

11) Page 11, “The interview’s mean is 27’51’’ (range: 18’50’’-52’51’’)”. I think authors used “´” instead of commas. Please revise.

12) Page 10, “(in-house prot. n. 2020/0013503 del 03/02/2020)”. “Del” is still in Italian, I think.

13) Page 10, “It was explained to the participants, who were assured they could discontinue participation without consequences”. It is unclear what that “it” is referred to. Please revise or just leave the main sentence “Participants were assured they could discontinue participation without consequences”.

14) Page 10, “Study population”. I think that “Study sample” or “Participants” would be a more appropriate title, as you are here presenting those who participated in the study and not the inclusion or exclusion criteria solely.

6. PLOS authors have the option to publish the peer review history of their article (what does this mean?). If published, this will include your full peer review and any attached files.

Reviewer #1: No

Reviewer #2: **Yes: **Marcella Bianchi

---

## [Author Response · Author response to Decision Letter 0]

13 Dec 2024

RESPONSE to REVIEWERS

Manuscript ID: PONE-D-24-38765

Title: Self-blaming as a Barrier to Lung Cancer Screening and Smoking Cessation Programs in Italy: A Qualitative Study

Journal: PLOS ONE

Dear Editor,

 I am sharing the article entitled: “Self-blaming as a Barrier to Lung Cancer Screening and Smoking Cessation Programs in Italy: A Qualitative Study” that we are re-submitting for publication in PLOS ONE.

In this cover letter, we explain the revisions made. Changes are tracked within the manuscript. We have discussed the comments with the co-authors and revised the text accordingly.

We thank the Editor and the reviewers for their time, careful review, and constructive feedback.

We hope we have met the expectations and are willing to provide further clarification as needed.

EDITORIAL COMMENTS

I have examined the paper myself and was able to secure the services of 2 expert reviewers. As you can see, they largely agree on the strengths of the paper and I concur. Reviewer 2 has provided an extensive list of useful suggestions, including a set of points that need to be referenced with which I strongly concur (e.g., Globocan will provide the most current incidence and prevalence figures for Italy etc).

RESPONSE: Thank you for your review and help with our manuscript!

My suggestions are as follows:

1. Please address each of the reviewers’ points in turn. I will add some further comments on some of these below, as well as comments of my own.

2. The abstract is definitely too long. The limit for PlosOne is 300 words and you are at about 430. This needs to be trimmed and certainly can be trimmed.

RESPONSE: We shortened the abstract to align it to journal’s requirements.

3. Please go through and address the paragraph structure of the manuscript. Avoid any single sentence paragraphs. These must be integrated with the paragraphs below or above, as appropriate, or inserted elsewhere.

RESPONSE: Thank you. We managed to avoid single sentence paragraphs where appropriate.

4. As you can probably tell by my name, I too can see the mixed English/Italian idiom. I strongly suggest you remove this entirely. In general, second language (L2) speakers have difficulty in comprehending idioms and so these should be avoided altogether in scientific and scholarly papers. That is, I would also suggest you remove “the straw the broke the camel’s back”.

RESPONSE: Thank you. We agreed and changed accordingly.

5. The sample size needs to be introduced carefully because it also points back to a process that is very important in qualitative research that has not been addressed in your manuscript, that of saturation. The reviewers ask why you chose 15 people. In general, in interview studies such as yours, the researchers would commence gathering interview data and only stop once each it was evident that no new material was forthcoming. This is called the point of saturation. It could occur after 5 or 25 or more interviews. It is difficult to reconcile how you would specify 15 people before you began interviewing and then have a sense of saturation. It is OK if you needed to, for pragmatic reasons, limit the number of interviews. But you need to explain that clearly and also to discuss what approaches you took to assess saturation in the circumstances.

RESPONSE: Thank you for raising this important point! We have now provided a more detailed explanation of the rationale behind the sample size/type and its recruitment in the manuscript, incorporating the relevant citations as requested by the reviewer.

As qualitative methodologist, I traditionally relied on the concept of data or thematic saturation to determine sample size, where saturation is understood as the point when no new themes or information emerge. This requires concurrent data collection and analysis or deep analysis of themes that become “representative” of the data collected. However, after engaging with recent debates surrounding saturation, I have reassessed this approach.

In particular, Braun and Clarke, whose guidelines we followed for the analysis, have questioned the use of data saturation in thematic analysis (TA). They emphasize that while data saturation is a widely cited concept, its operationalization often leads to an oversimplified or misleading view of the process. Specifically, they argue that the assumption of ‘information redundancy’ associated with saturation does not align with the values of reflexive thematic analysis. For reflexive TA, meaning is not simply extracted from data but generated through interpretation. As such, judgments about how many interviews or focus groups are enough cannot be standardized or predetermined, as they are inherently subjective and context-dependent.

Furthermore, Tight (2024) has critiqued the concept of saturation from another angle, particularly its use in health research. He notes that the idea of data saturation assumes that data collection should stop once only repetitive information is identified. However, he points out that since each life is unique, data can never truly be saturated—there are always new things to explore. Saturation, in this context, can also be seen as an acknowledgment of the researcher’s capacity to manage the data within a given timeframe. Tight warns against the tendency to "quantify" qualitative research by linking it to specific sample sizes, as this reduces the richness and flexibility of qualitative inquiry to its lowest common denominator. He advocates for a more open-ended approach, where decisions about stopping data collection are driven by criteria like "good enough" or "sufficiently well-developed" rather than arbitrary notions of saturation.

In light of these perspectives, we chose not to rely on saturation as a guiding principle for determining sample size. Instead, we opted for a purposive sample that we felt would provide the diverse perspectives necessary to answer our research question. This approach is consistent with the idea that meaning in qualitative research is constructed through interpretation, not simply extracted from data, and that the process of determining when to stop collecting data is inherently subjective and context-dependent. By focusing on theoretical sufficiency and conceptual depth, we believe our approach remains flexible and aligned with the values of reflexive thematic analysis.

We did not insert this explanation in the manuscript which could bring the readers far from the core messages. However, we are open to discuss how to integrate it or to differently justify our methodological choices.

• Braun, V., & Clarke, V. (2019). To saturate or not to saturate? Questioning data saturation as a useful concept for thematic analysis and sample-size rationales. Qualitative Research in Sport, Exercise and Health, 13(2), 201–216. https://doi.org/10.1080/2159676X.2019.1704846

• Tight, M. (2024). Saturation: An Overworked and Misunderstood Concept? Qualitative Inquiry, 30(7), 577-583. https://doi.org/10.1177/10778004231183948

6. For the TA, you should present a table with all the themes and an illustrative quote for each. This is in addition to the material in the body and will greatly help readers navigate your model.

RESPONSE: Thank you! We added a table with themes and a selection of meaningful quotations. We moved in some from the text and translated/inserted some others in addition.

7. Further, I think that the Discussion section misses the opportunity to relate cultural factors to psychological perceptions of screening, cancer, and lung cancer. In my current position in Hong Kong, I have been exposed to some distinctly Chinese perspectives on cancer that has contrasted with results of my previous work with lung cancer patients in Australia. Similarly, you might consider what aspects of the Italian sample and their interpretations can be drawn out. Much of the scholarship in this area is dominated by the Anglophone world and it is important to give voice to true cultural factors. For example, the potential for Italian fatalism seemed to be evident in some of the quotes I saw.

RESPONSE: Thank you for this valuable comment. We agree with your point and have revised the discussion, emphasizing the unique aspects at the beginning of the section.

8. As for the references to the Youtube video and other public health activities based on this study, these should be removed from the manuscript. They are certainly laudable, but it is up to the organization to promote these and to derive impact from this research. This does not belong in the Strengths and Limitations section, nor in any other part of the paper.

RESPONSE: Thank you. We agreed and deleted the parts.

Finally, you need to apply a qualitative assessment framework such as COREQ and incorporate it into the MS, most likely as an Appendix. This is very important for readers to be able to contextualise your work

RESPONSE: We agreed and added the COREQ as appendix.

Journal Requirements:

RESPONSE: We followed the template as suggested.

 This study contributed, without receiving funding, to the Italian Ministry of Health Pilot project “Progetto Pilota di un programma di screening per il tumore polmonare integrato con la cessazione del fumo: percorsi, selezione dei soggetti e protocolli diagnostici, in vista di una valutazione HTA”. The study was partially funded by Emilia Romagna Regional Health Authority DGR n° 1800/2020.

The research project “Analisi dei meccanismi d’azione psico-comportamentali con cui la proposta di adesione alla TAC spirale agisce sulle abitudini tabagiche e sulla disassuefazione al fumo” has been supported by a ‘5x1000’ national grant from the Italian League against Cancer (LILT-Rome).

This study was partially supported by the Italian Ministry of Health – Ricerca Corrente Annual Program 2025. 

RESPONSE: Thank you. We clarified the role in the new version of the cover letter.

This study contributed, without receiving funding, to the Italian Ministry of Health Pilot project

“Progetto Pilota di un programma di screening per il tumore polmonare integrato con la cessazione

del fumo: percorsi, selezione dei soggetti e protocolli diagnostici, in vista di una valutazione HTA”.

The study was partially funded by Emilia Romagna Regional Health Authority DGR n° 1800/2020.

The research project “Analisi dei meccanismi d’azione psico-comportamentali con cui la proposta

di adesione alla TAC spirale agisce sulle abitudini tabagiche e sulla disassuefazione al fumo” has

been supported by a ‘5x1000’ national grant from the Italian League against Cancer (LILT-Rome).

This study was partially supported by the Italian Ministry of Health – Ricerca Corrente Annual

Program 2025.

 This study contributed, without receiving funding, to the Italian Ministry of Health Pilot project “Progetto Pilota di un programma di screening per il tumore polmonare integrato con la cessazione del fumo: percorsi, selezione dei soggetti e protocolli diagnostici, in vista di una valutazione HTA”. The study was partially funded by Emilia Romagna Regional Health Authority DGR n° 1800/2020.

The research project “Analisi dei meccanismi d’azione psico-comportamentali con cui la proposta di adesione alla TAC spirale agisce sulle abitudini tabagiche e sulla disassuefazione al fumo” has been supported by a ‘5x1000’ national grant from the Italian League against Cancer (LILT-Rome).

This study was partially supported by the Italian Ministry of Health – Ricerca Corrente Annual Program 2025.

RESPONSE: Thank you for this clarification. We deleted the funding statements from the Acknowledgments Section. The Funding Statement is correct as it was provided. Thank you. We clarified it in the new version of the cover letter.

4. In the online submission form, you indicated that The data that support the findings of this study are available on request from the corresponding author, LG. 

RESPONSE: Thank you for your request regarding data availability. The qualitative data underlying the findings of this study consist of interview and focus group transcripts, which inherently contain personal and sensitive information. These transcripts were collected in accordance with ethical guidelines and research protocols approved by the ethics committee. Participants did not consent to the public sharing of their data, and public deposition would compromise their privacy and confidentiality, particularly since the data includes contextual information that could indirectly identify participants.

Furthermore, the data are in Italian, which may limit accessibility to an international audience. However, all key data and information necessary to understand the findings of this study are included within the manuscript itself. This includes the thematic analysis and illustrative quotes that provide transparency and rigor to the findings.

We are happy to provide further anonymized details or summaries of the data upon reasonable request, ensuring that confidentiality is maintained and ethical guidelines are upheld. As stated in the submission form, researchers may contact the corresponding author (LG) for inquiries regarding the data.

We kindly request an exemption from the data-sharing policy on these ethical grounds and look forward to your confirmation. 

Thank you for your understanding.

RESPONSE: Thank you for this note. We have been using a reference manager software for it.

No retracted articles has been included.

REVIEWER 1

Reviewer #1: Overall, the authors present a rigorous qualitative study that explores the perspectives of stakeholders involved in LCS and individuals eligible for the smoking cessation program. They se

---

## [Decision Letter · Decision Letter 1]

22 Jan 2025

Self-blaming as a Barrier to Lung Cancer Screening and Smoking Cessation Programs in Italy: A Qualitative Study

PONE-D-24-38765R1

Dear Dr. Ghirotto,

We’re pleased to inform you that your manuscript has been judged scientifically suitable for publication and will be formally accepted for publication once it meets all outstanding technical requirements.

Kind regards,

Enkeleint A. Mechili

Academic Editor

PLOS ONE

Additional Editor Comments (optional):

Reviewers' comments:

Reviewer's Responses to Questions

**Comments to the Author**

1. If the authors have adequately addressed your comments raised in a previous round of review and you feel that this manuscript is now acceptable for publication, you may indicate that here to bypass the “Comments to the Author” section, enter your conflict of interest statement in the “Confidential to Editor” section, and submit your "Accept" recommendation.

Reviewer #1: All comments have been addressed

Reviewer #2: All comments have been addressed

2. Is the manuscript technically sound, and do the data support the conclusions?

Reviewer #1: Yes

Reviewer #2: Yes

3. Has the statistical analysis been performed appropriately and rigorously? 

Reviewer #1: Yes

Reviewer #2: N/A

4. Have the authors made all data underlying the findings in their manuscript fully available?

Reviewer #1: No

Reviewer #2: No

5. Is the manuscript presented in an intelligible fashion and written in standard English?

Reviewer #1: Yes

Reviewer #2: Yes

6. Review Comments to the Author

Reviewer #1: The authors have addressed my comments well and I am particularly pleased to see the paper's valuable cultural contributions. I appreciate the work you have put into enhancing its quality. Thank you for your thoughtful revisions.

Reviewer #2: The authors have successfully incorporated the requested revisions throughout the manuscript, which, in my opinion, significantly improved its already high overall quality.

Here are my very last suggestions, which are, of course, optional:

• Regarding “ex” cigarette users, I would suggest using the word “former” rather than “earlier,” as it sounds more appropriate to me (but English is not my mother tongue, so please take this with a grain of salt).

• I apologize for not catching the meaning of the ´´ symbols to indicate minutes earlier. However, even in the current form, you could consider making it even more explicit what you are referring to (e.g., “On average, the interviews lasted approximately 27 minutes (ranging from 18 to 52 minutes)”).

• Thank you for adding a proper description of the sample composition and its rationale. I am accustomed to referring to the concept of data saturation when defining sample sizes. While I do believe that the new version of the manuscript makes it sufficiently clear how choices were made, mentioning the arbitrary choice of recruiting 15 participants still seems somewhat strange to me. However, I particularly appreciate the reflection on the material unavailability of the former ITALUNG participants, which, in my opinion, helps justify the decision to set a relatively low minimum number (instead of a precise number of 15). Nevertheless, I also think the current version contains all the information necessary for readers to form their own opinions on the matter and interpret the results with appropriate caution. Of course, the overall merits of the manuscript are still sufficient to recommend its publication. Therefore, I would defer to the editor’s expertise for the final decision on this specific issue.

7. PLOS authors have the option to publish the peer review history of their article (what does this mean?). If published, this will include your full peer review and any attached files.

Reviewer #1: No

Reviewer #2: **Yes: **Marcella Bianchi

---

## [Editor Report · Acceptance letter]

31 Jan 2025

PONE-D-24-38765R1 

PLOS ONE

Dear Dr. Ghirotto, 

I'm pleased to inform you that your manuscript has been deemed suitable for publication in PLOS ONE. Congratulations! Your manuscript is now being handed over to our production team.

Kind regards, 

on behalf of

Prof. Assoc. Dr. Enkeleint A. Mechili 

Academic Editor

PLOS ONE